# Aminoglycosides in the Intensive Care Unit: What Is New in Population PK Modeling?

**DOI:** 10.3390/antibiotics10050507

**Published:** 2021-04-29

**Authors:** Alexandre Duong, Chantale Simard, Yi Le Wang, David Williamson, Amélie Marsot

**Affiliations:** 1Faculté de Pharmacie, Université de Montréal, Montréal, QC H3T 1J4, Canada; yi.le.wang@umontreal.ca (Y.L.W.); david.williamson@umontreal.ca (D.W.); amelie.marsot@umontreal.ca (A.M.); 2Laboratoire de Suivi Thérapeutique Pharmacologique et Pharmacocinétique, Faculté de Pharmacie, Université de Montréal, Montréal, QC H3T 1J4, Canada; 3Faculté de Pharmacie, Université Laval, Québec, QC G1V 0A6, Canada; chantale.simard@pha.ulaval.ca; 4Centre de Recherche, Institut Universitaire de Cardiologie et Pneumologie de Québec, Québec, QC G1V 4G5, Canada; 5Hôpital Sacré-Cœur de Montréal, Montréal, QC H4J 1C5, Canada; 6Centre de Recherche, CHU Sainte Justine, Montréal, QC H3T 1C5, Canada

**Keywords:** aminoglycosides, population pharmacokinetic modeling, intensive care unit, critically ill

## Abstract

Background: Although aminoglycosides are often used as treatment for Gram-negative infections, optimal dosing regimens remain unclear, especially in ICU patients. This is due to a large between- and within-subject variability in the aminoglycoside pharmacokinetics in this population. Objective: This review provides comprehensive data on the pharmacokinetics of aminoglycosides in patients hospitalized in the ICU by summarizing all published PopPK models in ICU patients for amikacin, gentamicin, and tobramycin. The objective was to determine the presence of a consensus on the structural model used, significant covariates included, and therapeutic targets considered during dosing regimen simulations. Method: A literature search was conducted in the Medline/PubMed database, using the terms: ‘amikacin’, ‘gentamicin’, ‘tobramycin’, ‘pharmacokinetic(s)’, ‘nonlinear mixed effect’, ‘population’, ‘intensive care’, and ‘critically ill’. Results: Nineteen articles were retained where amikacin, gentamicin, and tobramycin pharmacokinetics were described in six, 11, and five models, respectively. A two-compartment model was used to describe amikacin and tobramycin pharmacokinetics, whereas a one-compartment model majorly described gentamicin pharmacokinetics. The most recurrent significant covariates were renal clearance and bodyweight. Across all aminoglycosides, mean interindividual variability in clearance and volume of distribution were 41.6% and 22.0%, respectively. A common consensus for an optimal dosing regimen for each aminoglycoside was not reached. Conclusions: This review showed models developed for amikacin, from 2015 until now, and for gentamicin and tobramycin from the past decades. Despite the growing challenges of external evaluation, the latter should be more considered during model development. Further research including new covariates, additional simulated dosing regimens, and external validation should be considered to better understand aminoglycoside pharmacokinetics in ICU patients.

## 1. Introduction

Aminoglycosides are a class of antibiotics used as treatment for Gram-negative infections in patients hospitalized in intensive care units (ICUs). Life-threatening infections, often caused by Gram-negative bacteria [1,2], may lead to pathophysiological conditions, such as sepsis, influencing the pharmacokinetics (PK) of many drugs including antibiotics [3]. For example, ICU patients may exhibit an increased volume of distribution, causing lower aminoglycosides peak concentrations [4]. Therefore, the selection of both the appropriate antimicrobial therapy and its respective dosage are essential for clinical cure [5]. As aminoglycosides follow concentration-dependent pharmacodynamics, the achievement of a peak concentration (C_max_) over minimum inhibitory concentration (MIC) ratio greater than 10 is warranted for a clinical response [6]. Although the C_max_/MIC target is primarily used in clinical situations due to its simplicity, multiple studies have shown that an area under the curve (AUC) to MIC ratio greater than 80–100 is the better pharmacokinetic/pharmacodynamic (PK/PD) indicator for efficacy [6,7,8]. Considering the narrow therapeutic index of aminoglycosides with potential nephrotoxicity and ototoxicity, therapeutic drug monitoring (TDM) has been used to achieve these targets while minimizing toxicity by individualizing treatments [9]. This practice is especially crucial in ICU patients that suffer from septic shock where the survival rate is increased with the timely administration of an appropriate antibiotic [10].

In recent years, antibiotic dosing regimens have been developed with the help of population pharmacokinetic (PopPK) modeling and simulation [11]. Multiple studies have established PopPK models to characterize PK parameters and to gain a better understanding of the variability of aminoglycoside clinical response based on ICU patients’ characteristics. These studies have used nonlinear mixed effects modeling to target and quantify the contribution of specific demographic and pathophysiological characteristics that may influence the aminoglycoside PK profile. This modeling method has been considered as one of the principal approaches in PopPK modeling due to the possibility of having sparse data for each subject while evaluating residual and interindividual variabilities [12]. Moreover, PopPK models can also be used to develop dosing recommendations by simulating several dosing regimens based on different PK/PD targets. However, it is also important to assess the validity of these models and the efficacy of the dosing recommendations in actual clinical settings in large populations. Generally, clinical pharmacokinetic studies must present several key items to better ensure transparency in the reporting of the results [13].

The aim of this review was to provide comprehensive data on the pharmacokinetics of aminoglycosides in patients hospitalized in ICU by summarizing all published PopPK models in ICU patients for amikacin, gentamicin, and tobramycin.

## 2. Data Sources

### 2.1. Search Strategy

A literature search was conducted in the Medline/PubMed database, from its inception until March 2020, using the following terms: (*amikacin* OR *gentamicin* OR *tobramycin)* AND [(*pharmacokinetics*/or *renal elimination*/) OR (*pharmacokinetic** OR ((*pharmaco* OR *drug*) ADJ *kinetic**) OR *area under curve*? OR *AUC* OR (*renal* ADJ (*elimination*? or *excretion*? or *clearance*?))) OR (((*nonlinear* OR *non-linear*) ADJ *mixed effect model**) OR *NONMEM* OR *WinNonMix* OR *P-PHARM* OR *NLMIXED* OR *ADAPT*)] AND (EXP *population*/OR *population groups*/OR (*population*? OR *ethnic group*?)) AND [*critical care*/OR *intensive care* or EXP *intensive care units*/OR *critical illness*/OR ((*intensive* OR *critical*) ADJ *care*?) OR *ICU* OR ((*respiratory* OR *coronary*) ADJ *care unit*?) OR (*critical** ADJ (*ill* OR *illness*? OR *disease*?))]. Additional relevant studies were manually screened from the reference list of selected articles. The phases of systematic review are displayed in a flowchart (Figure 1), as described by the PRISMA 2009 statement for reporting systematic reviews and meta-analyses [14]. The research strategy was completed by two authors, and cross-verification was performed.

### 2.2. Inclusion Criteria

Eligible studies had to meet the following inclusion criteria: (1) the article described a population pharmacokinetic model; (2) the treatment was intravenous amikacin, gentamicin, or tobramycin; (3) the studied population consisted of ICU adult patients; (4) the article was published in the English language.

### 2.3. Exclusion Criteria

We excluded articles from this review if they met one of the following criteria: (1) a noncompartmental approach was used; (2) the studied population was composed of only cystic fibrosis patients; (3) the studies were published before 2015 for amikacin (this review served as an update to the amikacin review by Marsot et al. [15]; (4) they were review articles.

### 2.4. Data Extraction

The following information was extracted from relevant articles: first author, year of publication, population characteristics (number of males and females, age, bodyweight, height, and body mass index), study design, dosage regimen, sample collection (samples per patient, total samples, and sample frequency), population PK modeling methods (software used, model and evaluation method used), the formula of PopPK structural and statistical models, PK parameters, and tested and retained covariates. The model evaluation methods were divided into basic internal (goodness-of-fit plots), advanced internal (bootstrap resampling, Monte Carlo simulations, visual predictive check, normalized prediction distribution error, etc.), and external evaluation. This step was done by two authors, and cross-verification was performed to ensure the accuracy of information extracted. Data extraction was based on the several items presented in the checklist created by *ClinPK* [13], as per Appendix A.

## 3. Data Analysis

### 3.1. Study Selection

A total of 78 studies were identified through the Medline/PubMed database, of which there were 26 articles for amikacin, 38 for gentamicin, and 14 for tobramycin. After assessing the articles for eligibility by applying the inclusion and exclusion criteria, 19 publications were selected. In total, six, 11, and five PopPK models were analyzed for amikacin [16,17,18,19,20,21], gentamicin [21,22,23,24,25,26,27,28,29,30,31], and tobramycin [32,33,34], respectively (Figure 1).

### 3.2. Population Characteristics

The characteristics of the population studies are presented in Table 1. The mean population age from these studies ranged from 32 years [34] to 74 years [31] with the mean bodyweight ranging from 51 kg [25] to 92.5 kg [27].

### 3.3. Study Designs and Protocols

In Table 1, among the 19 publications across all three aminoglycosides, the numbers of retrospective and prospective designs were similar, with 10 and eight, respectively. Another study had both retrospective and prospective designs [23]. Patients were mostly administered aminoglycosides through intravenous infusion with only two studies including intravenous bolus administration. The number of patients included ranged from 14 [27] to 208 [34]. Furthermore, seven studies included fewer than 30 patients in their PopPK analysis [17,20,21,27,28,31]. The number of total samples and blood samples collected per patient varied across all studies for all three aminoglycosides. Peak and trough samples were usually the samples collected for studies following a TDM protocol (*n* = 14), whereas a complete PK profile of the aminoglycoside was required for PK studies (*n* = 5).

Amikacin was mostly administered following a once-daily dosing regimen in six respective study protocols, except for one where it was unknown, but it was mentioned that the dosing regimen followed establishment’s standards [18]. For amikacin, the actual doses administered to the study populations ranged from 23 mg/kg/day to 41 mg/kg/day. Similarly, gentamicin dosing regimens were mostly once-daily administration. One prospective study administered three different dosing intervals to their study population: once-daily, twice-daily, and thrice-daily [25], whereas another prospective study administered five different dosing intervals ranging from twice-daily to once every 3 days [30]. For all gentamicin studies, the daily dosage regimens, as well as the actual administered doses, were similar, ranging from 3 mg/kg to 7 mg/kg. Similarly, tobramycin was also given following a once-daily administration with dosing regimens and actual administered doses ranging from 5 mg/kg/day to 7 mg/kg/day.

### 3.4. Population Pharmacokinetic Analysis

All 19 studies included in this review used nonlinear mixed effect methods to analyze their data and develop PopPK models. As per Table 2, a version of NONMEM software was used for the modeling in more than half of the studies (*n* = 10) [19,22,23,24,25,26,27,32,33,34]. Other software used included NPAG, a function from the software Pmetrics (*n* = 2), and the NPEM software (*n* = 2). For model evaluation, more than half of these studies only used advanced internal evaluation, such as the bootstrap resampling method (*n* = 10), while three studies used both advanced internal and/or external evaluation with several external subjects ranging from 13 to 32 [19,29,33]. Tobramycin pharmacokinetics was described by a two-compartment model (*n* = 3) [32,33,34], while amikacin and gentamicin pharmacokinetics were described by single-compartment (amikacin *n* = 1 [19], gentamicin *n* = 7 [23,24,25,28,29,30,31]) and two-compartment models (amikacin *n* = 5 [16,17,18,20,21], gentamicin *n* = 4 [21,22,26,27]).

### 3.5. Estimated Parameters

The mean estimated clearances (CL) were comparable across aminoglycosides, whereas the mean volume of distribution (Vd) was slightly higher in amikacin compared to gentamicin and tobramycin. As per Figure 2, the median values (range) of CL were 3.7 L/h (2.0–7.1 L/h), 3.0 L/h (1.15–5.7 L/h), and 3.95 L/h (3.14–7.23 L/h) across all studies for amikacin, gentamicin, and tobramycin, respectively, whereas the median values (range) of Vd were 34.9 L (20.3–46 L), 29 L (19–53 L), and 35 L (30–53 L) for amikacin, gentamicin, and tobramycin, respectively. CL and Vd values are also presented per study in Appendix A for single- and two-compartmental models, respectively.

### 3.6. Random Effect Modeling

Interindividual variability (IIV) for the main PK parameters was estimated only in one-third of the amikacin studies [18,19], whereas it was estimated in seven out of the 11 gentamicin studies [22,23,24,25,26,27,28]. For tobramycin, all five studies estimated IIV for both CL and Vd [24,28,32,33,34]. For amikacin, the median (range) values of IIV in CL and Vd (or central volume) following the inclusion of covariates were 47.0% (27.2–58.7%) and 33.6% (21.7–43.3%), respectively (*n* = 3 for each parameter) [18,19], with one study expressing IIV as ω^2^ (variance of eta) [18]. As for gentamicin, the median (range) values of IIV in CL and Vd (or central volume) following the inclusion of covariates were 47.0% (29.3–83.7%) and 17.2% (11.9–64.4%), respectively (*n* = 8 and 7 for CL and Vd, respectively) [24,28,32,33,34]. For tobramycin, the median (range) values of IIV in CL and Vd (or central volume) following the inclusion of covariates were 30.8% (25.9–83.7%) and 15.2% (3–64.4%), respectively (*n* = 5 for each parameter) [24,28,32,33,34]. However, the highest IIV values for both CL and Vd were taken from a study that collected both gentamicin and tobramycin samples in their study population [24].

Across all aminoglycosides, the studies tested additive (*n* = 2) [19,28], proportional (*n* = 6) [18,22,27,32,33,34], or mixed error (additive and proportional) (*n* = 5) [20,23,24,25,26] models in order to determine residual variability. As per Appendix A, for amikacin, residual variability was estimated using a proportional model (*n* = 1) [18], an additive model (*n* = 1) [19], and a mixed model (*n* = 1) [20]. As for gentamicin, the median (range) residual variability using a proportional model was 27.3% (20.8–33.8) (*n* = 2) [22,27], whereas the residual variability was estimated using an additive model in a single study where both gentamicin and tobramycin samples were used in the model development [28]. The medians (ranges) using a mixed model were 24.3% (19.4–32%) and 0.056 mg/L (3.81 × 10^−4^ mg/L–0.13 mg/L) (*n* = 3) [24,25,26]. Another study presented the residual variability estimated with a mixed model as variance [23]. For tobramycin, the median (range) residual variability using a proportional model was 21% (20.4–23.7%) (*n* = 3) [32,33,34].

### 3.7. Inclusion of Covariates

Appendix A summarizes the tested and significant covariates. For estimated clearance (CL), the most common retained covariate was creatinine clearance calculated using the Cockcroft–Gault (CG) equation (*n* = 8) [16,18,19,20,23,25,32,33]. Moreover, multiple covariates related to weight (total bodyweight (TBW) [17,29], ideal bodyweight (IBW) [22], and lean bodyweight [27]) and body size (height [32] and free fat mass [34]) were also included (*n* = 1, for each). Other retained covariates for CL were glomerular filtration rate [24], sex, serum creatinine, age [34], usage of renal replacement therapy (intermittent hemodialysis [23] or continuous venovenous hemofiltration (CVVH) [22]), and the inverse of the final plasma creatinine concentration recorded in µmol/L before commencement of extended daily diafiltration (EDD-f) [27]. For the estimated Vd, most common retained covariates were related to weight and body size (body surface area (*n* = 1) [16], adjusted bodyweight (*n* = 1) [18], bodyweight (*n* = 1) [24], ideal bodyweight (*n* = 1) [22], and free fat mass (*n* = 1) [34]). Other retained covariates for Vd were albumin [22] and sex [34] (*n* = 1 each).

### 3.8. Simulation of Dosing Regimens

As per Table 2, amongst the 19 articles selected in this review for all three aminoglycosides, 12 (amikacin (*n* = 4), gentamicin (*n* = 5), and tobramycin (*n* = 3)) of them simulated optimal dosing regimens in their respective population with various therapeutic targets [16,17,18,19,23,24,25,26,27,32,33,34]. All 12 studies included at least a target related to C_max_, while half of them also included a target related to AUC_0–24_ or AUC_0–48_, and five studies added trough concentration as one of their therapeutic or toxicity targets. Generally, dosing regimens simulated across studies were similar for all three aminoglycosides, with some adjustments based on the populations’ characteristics. Many studies used various targets for their simulations. For amikacin, principal PK/PD targets were C_max_/MIC ≥ 8, AUC_0–24_/MIC ≥ 75, and C_min_ ≤ 2.5 mg/L [16,17,18]. For gentamicin, main PK/PD targets were C_max_/MIC between 8 and 10, considering an MIC ranging from 1 to 2 mg/L [23,24,25,26,27]. As for tobramycin, C_max_ values were targeted to be within 6 mg/L and 20 mg/L considering an MIC of 1 to 2 mg/L and C_min_ values were set to be ≤1 mg/L [32,33,34].

## 4. Discussion

To treat severe infections, the administration of aminoglycosides in special populations has led to an increase in interest in aminoglycoside pharmacokinetics. Noticeably, a considerable number of PopPK models have been developed for ICU patients in the last decade [16,17,18,19,20,22,25,26,27,29,32,34]. The 19 articles presented in this review exhibit many resemblances but also differences in the covariates included, the structure of the model, and the simulation of dosing regimens. Studies presenting a design with TDM samples or a sparse sampling schedule were mostly associated with single-compartment models (*n* = 8), whereas full-profile sampling partially led to two-compartment models (*n* = 11). In fact, Marsot et al. suggested in their review that single-compartment models could lead to an inaccurate estimation of aminoglycoside Vd [15]. Although median CL and Vd values were comparable across aminoglycosides, as shown in Figure 2, the parameter values tended to vary from one study to another for each drug. As described previously, ICU patients are prone to present additional comorbidities, such as cardiovascular dysfunction, sepsis, burns, or use of vasopressors, and/or develop complications, such as acute kidney injury (AKI) or, conversely, augmented renal clearance (ARC). Although ARC is expected to being present in 20–65% of critically ill patients [35], it was only considered in a few studies in this review [16,18,19,25]. These complications usually lead to divergence in PK values as compared to healthy patients [36]. As per Figure 2a, based on a similar dosing regimen, median CL values for all three drugs in this present study were generally lower as compared to values in healthy volunteers: 6.48 L/h, 4.03 L/h, and 7.02 L/h for amikacin, gentamicin, and tobramycin, respectively [37,38,39,40]. As shown in Figure 2b, the median Vd values for all three drugs in this review were higher than values shown in healthy volunteers: 16.15 L, 13.3 L/70 kg, and 20 L/70 kg for amikacin, gentamicin, and tobramycin, respectively [37,38,39,40].

### 4.1. Major Covariates

In addition of the changes due to critical illness, ICU patients may present other physiological characteristics potentially impacting aminoglycoside pharmacokinetics. To better understand the inter- and intra-variability of aminoglycosides pharmacokinetics, the following covariates were the most retained in PopPK models: bodyweight (*n* = 7) and renal clearance (*n* = 8).

#### 4.1.1. Renal Function

Among the 12 studies with normal renal function patients that performed a covariate analysis, seven studies included CL_CR_ calculated using the Cockcroft–Gault equation (CL_CG_) in order to better estimate values of CL or Vd [16,18,19,23,25,32,33]. To illustrate the impact of CL_CR_ on aminoglycoside CL, we plotted aminoglycoside CL against this covariate according to the values and model equations reported by the studies that included CL_CR_ (Figure 3). This plot shows how differences in CL_CR_ caused important variations in aminoglycosides CL within the same study group. Considering that the CL_CG_ includes the age, total bodyweight, and sex of an individual, these variables are, therefore, also considered in the estimation of aminoglycoside CL or Vd.

Although CL_CG_ seems to be frequently used in guidelines [41], it might not represent the most accurate way of estimating aminoglycoside clearance [42]. In fact, CL_CG_ is known to overestimate the CL_CR_ in underweight individuals [43]. As for obese individuals, the usage of CL_CG_ with IBW tends to underestimate the CL_CR_, while the usage of TBW overestimates the CL_CR_ [43]. Many studies have suggested that CL_CG_ should not be used in intensive care settings [44,45,46,47]. Moreover, since CL_CR_ considers glomerular filtration, as well as tubular secretion [48], measurements of GFR have been suggested to be a more precise estimate of aminoglycoside clearance [49]. In fact, the aminoglycoside elimination pathway mainly involves glomerular filtration, while tubular secretion and reabsorption are minimal, even when GFR levels are low. Zarowitz et al. compared gentamicin and tobramycin clearances to inulin (GFR) and CL_CG_, and their results showed a better linear regression between inulin and GFR (*R^2^* = 0.93) compared to the linear regression between inulin and CL_CG_ (*R^2^* = 0.76) [49]. Moreover, Lim et al. also compared different estimators of GFR with the traditional CL_CG_, and they determined that the best predictor of aminoglycoside clearance would be the estimation of glomerular filtration rate by CKD-EPI adjusted for BSA [41]. Considering the high prevalence of CL_CG_ among the studies included in this review and its frequent usage in dosing guidelines, the better estimator between CL_CG_ and GFR, in terms of accuracy and efficacy in clinical settings, is still debatable.

Despite age not being a significant covariate in the estimation of aminoglycoside PK parameters in ICU patients, except when considered in the CG equation, advanced age is often associated with several physiological changes such as loss of kidney function and modifications in body composition influencing drug absorption and distribution of drugs [50]. In fact, it has been suggested that gentamicin renal clearance seemed to decline more significantly after reaching 60 to 70 years of age [51]. However, it was also mentioned that this decrease in gentamicin clearance might also be caused by other underlying diseases. The authors pointed out that the gentamicin Vd slightly varied across different ranges of age (39, 61, and 80 years old). Although age has been considered as an independent factor of nephrotoxicity and ototoxicity, several clinical studies mentioned that gentamicin clearance was influenced mainly by renal function and that the impact of age, by itself, is not significant [51,52,53].

#### 4.1.2. Bodyweight and Body Size

Since aminoglycosides are administered following a weight-based dose, the selection of the right weight parameter is essential to avoid overestimating or underestimating the dose needed. For example, in overweight patients, it is recommended to use an adjusted bodyweight that will consider a fraction of the excess bodyweight (total bodyweight–ideal bodyweight) [43]. Obesity is associated with major physiological changes such as an increased Vd for antibiotics, e.g., aminoglycosides [54]. Therefore, administration of higher doses to reach targeted serum concentrations is needed. In several studies presented in this review, patient weight was determined significant in the estimation of amikacin and gentamicin clearances (*n* = 3) [17,22,27] and volume of distribution (*n* = 3) [19,22,24]. To illustrate the impact of bodyweight in general on aminoglycoside Vd, the latter was plotted against this covariate according to the values and model equations reported by the studies that included a BW variable (Figure 3). Variations within BW from a same study seem to imply changes in aminoglycoside Vd. As mentioned earlier, bodyweight also has an influence on the estimation of the CL_CR_, especially if CL_CG_ is used. All seven studies that included CL_CG_ in their final PopPK model used TBW in the CG equation [16,18,19,23,25,32,33]. For studies that included impaired renal patients, each study retained a bodyweight parameter in one of the two parameters their final model [17,19,22,27]. Indeed, the inclusion of a bodyweight parameter is expected in this population considering that bodyweight is used in order to determine dialysate or ultrafiltration flow rate for renal replacement therapy (RRT) [17,22,23,27].

For body size parameters, only body surface area (BSA), lean body mass according to the equation of Chennavasin (LBMc), and free fat mass (FFM) were retained covariates in amikacin, gentamicin, and tobramycin models, respectively [16,29,34]. In fact, these three covariates were retained in the estimation of aminoglycoside Vd. Although BSA has rarely been mentioned as a covariate influencing aminoglycoside PK, it was suggested by Boidin et al. that the use of BSA might lower the risk of exposure in overweight patients [16,55]. In fact, BSA considers both the bodyweight and height, where the latter is much less variable than bodyweight in ICU adult patients [56]. Recent studies did suggest dose recommendations based on height (mg/cm) instead of bodyweight for tobramycin in cystic fibrosis patients [57,58].

Although the inclusion of parameters related to bodyweight or body size in the final model of most studies allowed a reduction in IIV, the latter remains relatively high across studies. This variability could be explained by the inaccuracy and variability of the estimation of TBW or actual bodyweight of ICU patients [59,60].

### 4.2. External Validation and Application

External validation is one of the strictest approaches in model testing and consists of applying a new dataset within a final model to determine the accuracy and reproducibility of the model and in which conditions it would be applicable. Different strategies and steps are possible in order to adequately evaluate models from the literature. For more information on these strategies, refer to the Appendix A.

In this review, most studies performed at least advanced internal validation (*n* = 13) but only three of them validated their model with another dataset [19,29,33], resulting in adequate bias and inaccuracy values. Although each of these three models was externally validated using data from independent patients, this does not imply that these models could be easily applied into other datasets from similar populations. Moreover, while external validation is highly preferred during model evaluation, the number of studies performing it is rather insufficient [61]. This lack of external validation could be due to the difficulty of collecting data from enough patients with similar characteristics from another ICU to build a high-quality validation dataset. Furthermore, external validation in antimicrobials is known to often lead to inadequate bias and inaccuracy values [62,63,64], thus suggesting that a certain challenge still remains.

The conception of a meta-model for each aminoglycoside may also be feasible by including the characteristics (covariates, error models, initial estimates) from the best-performing models following external validation with an independent dataset. The development of this meta-model is, therefore, derived from the independent dataset while also being based on previously published PK models.

### 4.3. Simulation of Dosing Regimens

Firstly, amikacin dosing recommendations in critically ill patients without RRT were simulated in two articles [16,19]. In Boidin et al., an optimal initial amikacin dose of 3.5 g showed a better PTA for C_max_ ≥ 64 mg/L and AUC_0–24_ ≥ 600 mg*h/L compared to the conventional 30 mg/kg of corrected bodyweight (CBW), considering an MIC of 8 mg/L [16]. It was suggested that an increase in the dosing interval up to 36 or 48 h might be feasible in critically ill patients with normal to moderate renal function. In fact, several recommendations were simulated on the basis of different values of the two significant covariates in their respective PopPK model, CL_CG_ (10 mL/min to 250 mL/min), and BSA (1.5 m^2^ to 2.5 m^2^). As for Aréchiga-Alvarado et al., different daily dosing recommendations were simulated on the basis of three different MICs (4 mg/L, 8 mg/L, and 16 mg/L) and CL_CR_ ranging from 60 mL/min to 200 mL/min [19]. Considering an MIC of 8 mg/L, a 30 mg/kg daily dose would be able to show a TAR over 80% and 75% for patients with CL_CR_ lower than 140 mL/min and greater than 140 mL/min, respectively. As for amikacin dosing recommendations in critically ill patients RRT, two studies showed similar results in terms of optimal dosing regimens. In fact, Roger et al. and Carrié et al. suggested, respectively, that a dose of 25 mg/kg every 48 h and a dose ranging from 25 mg/kg and 30 mg/kg every 36 to 48 h were the most appropriate in order to maximize TAR for C_max/_MIC ≥ 8 and AUC_0–24_ ≥ 70 or AUC_0–24_ ≥ 75 with an MIC of 8 mg/L [17,18].

Secondly, gentamicin and tobramycin dosing recommendations in critically ill patients without RRT were simulated in five different articles [24,25,32,33,34]. Three out of the five studies established similar dosing recommendations with an initial starting dose of 6 to 7 mg/kg or a daily dose of 7 mg/kg [24,25,26]. The other study from Conil et al. provided a graphical representation of TAR for C_max_ > 10 mg/L, C_trough at 24h_ < 1 mg/L, and AUC between 80 and 125 mg*h/L according to different fixed dose regimens [32]. Their main takeaway was that these targets were not reached simultaneously in more than 45% of patients. Furthermore, only half of the population was able to attain the target for AUC with daily fixed dosages of 375 and 400 mg. The other study from Aarons et al. simulated dosing regimens on the basis of CL_CR_ values [33]. All dosing regimens proposed were presented as a sequence: a fixed dose administered for the first 48 h with a dosing interval ranging from 8 h to 24 h depending on the CL_CR_. Following the first 48 h, a maintenance dose was to be administered as per the same dosing interval. The first period of 48 h was chosen according to the authors’ assumption that aminoglycoside concentration was to be detectable and, thus, have the possibility of dose adaptation [33]. As for patients under RRT, Teigen et al. demonstrated that, on the basis of PK/PD targets of C_max_ ≥ 8 mg/L and AUC_48_ between 140 and 24 0 mg·h/L, three fixed starting doses (300 mg, 240 mg, 220 mg) prior to dialysis are related to a better TAR compared to post-dialysis administration [23]. Furthermore, Roberts et al. showed that a dosing of gentamicin 6 mg/kg every 48 h and administered 30 min prior to RRT (EDD-f in this situation) was able to achieve PK/PD targets compared to daily 7 mg/kg administration [27].

Among the articles that performed simulation of dosing regimens, five of them simulated optimal dosing regimens interpolated from the actual dose administered in their respective study populations [17,18,24,25,26], whereas the other three resulted in optimal dosing regimens extrapolated from the actual dosing regimen administered [16,19,34]. Results from simulations based on inter- and extrapolation should be interpreted cautiously considering the high variability observed in the estimation of PK parameters for all aminoglycosides.

## 5. Conclusions

Although many PopPK models for aminoglycosides exist in the literature, important variability remains. Despite multiple covariates being tested across all studies, the significant covariates would still be creatinine clearance and bodyweight for aminoglycoside clearance and volume of distribution, respectively. Moreover, considering that aminoglycoside-induced toxicity is reported to be more frequent amongst individuals with mitochondrial DNA mutations, such as m.1555A>G and m.1494C>T in the 12S rRNA gene [65], pharmacogenetics should be taken into account in future PopPK models. Several limitations are to be considered; seven study populations had fewer than 30 subjects, and more than half of the articles had retrospective designs with few aminoglycoside samples.

Although simulations have been carried out and help us to suggest optimal dosages, it should not be forgotten that many models were not evaluated externally and, therefore, may not be generalizable. Moreover, these dosing regimens were taken from a small sample size of studies, and additional research on simulated dosing regimens based on specific subpopulations should be necessary to optimize aminoglycoside individualized dosing. TDM remains essential in the ICU population to achieve therapeutic success while minimizing the likelihood of toxicity.

## Figures and Tables

**Figure 1 antibiotics-10-00507-f001:**
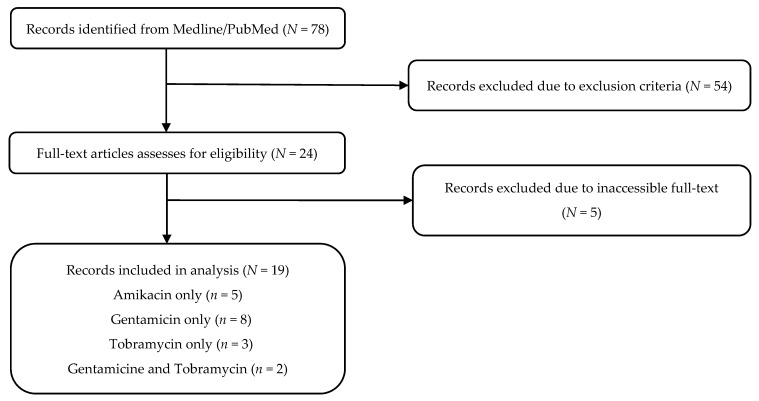
PRISMA chart.

**Figure 2 antibiotics-10-00507-f002:**
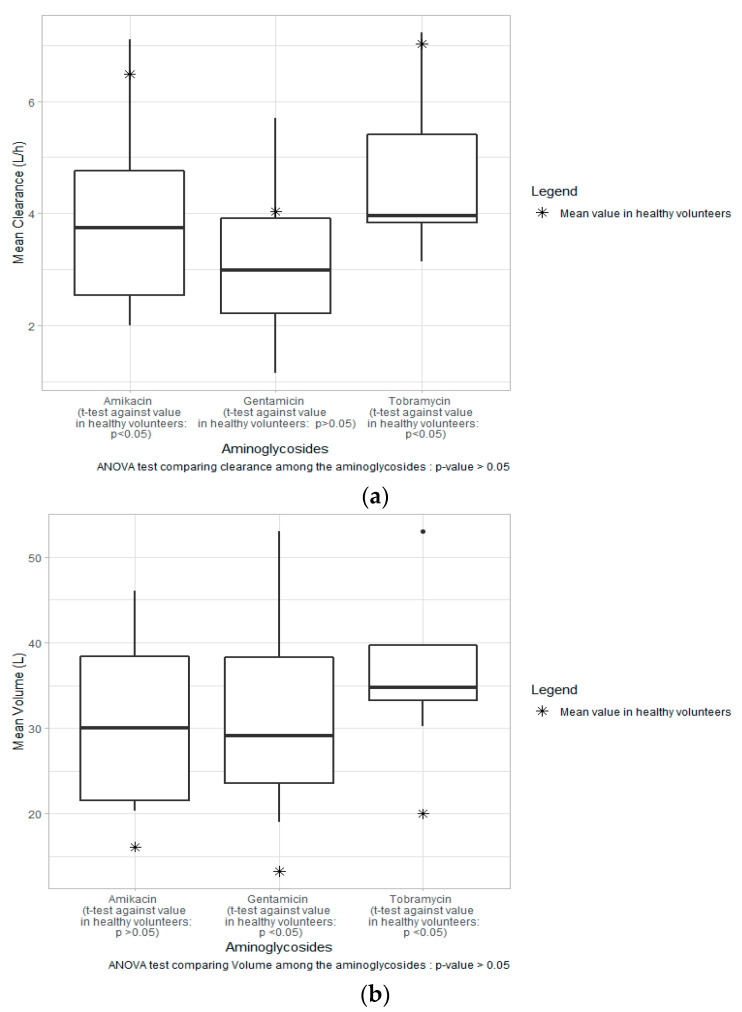
(**a**) Range of mean clearance across studies stratified by aminoglycosides (amikacin, gentamicin, and tobramycin) with mean clearance value in healthy volunteers (dotted line). (**b**) Range of mean volume of distribution across studies stratified by aminoglycosides (amikacin, gentamicin, and tobramycin) with mean volume of distribution value in healthy volunteers (dotted line).

**Figure 3 antibiotics-10-00507-f003:**
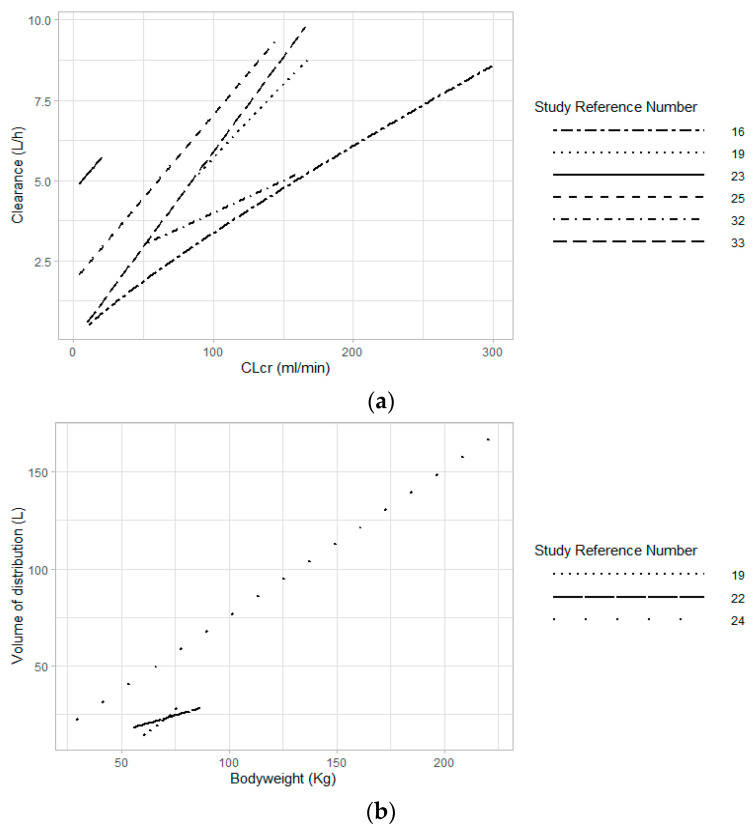
(**a**) Aminoglycoside clearance values against range of creatinine clearance in the respective studies. (**b**) Aminoglycoside volume of distribution values against range of bodyweight in the respective studies. Note: Two studies used IBW [19,26] and one used TBW [24] in their model.

**Table 1 antibiotics-10-00507-t001:** Summary of patients’ demographics and clinical protocol for all population pharmacokinetic studies included in this review for amikacin, gentamicin, and tobramycin.

Drug	Study	Year	Study Type	Population						Aminoglycoside Administration		Samples		
				Patient Characteristics	*N* (Male/Female)	Age (Years) ^a^	Body Weight (kg) ^a^	Height (cm) ^a^	BMI (kg/m^2^) ^a^	Dosage Regimen	Administered Dose (mg/kg) ^a^	Samples per Patient	Total Samples	Sample Frequency (h)
Amikacin	Boidin C [16]	2019	Retrospective (TDM)	Critically ill with sepsis	166 (108/58)	65 (19–85) ^b^	76.5 (41.5–137.5) ^b^	170 (137–190) ^b^	25.6 (16–46) ^b^	Administered Daily	23.4 (11–39.7) [20.0–27.0] ^b^	NR	395	Peak and trough
	Roger C [17]	2016	Observational pharmacokinetic study	Critically ill undergoing CVVH (*n* = 9) and CVVHDF (*n* = 11)	16 (12/4)	72 [65–75] ^b^	80 [73–89] ^b^	167 [162–178] ^b^	27 [24–32] ^b^	15–30 mg/kg every 24 or 36 h	NR	9	261	Predose, end of infusion (0.5), 1,1.5, 2, 4, 8, 12, and 24
	Carrié C [18]	2020	Retrospective (TDM)	Critically ill septic patients treated by OA/NPT	70 (53/17)	65 [51–73] ^b^	80 [65–94] ^b^	NR	27 [25–32] ^b^	As per medical care by the local Department of Laboratory Medicine	26 [24–29] ^b^	NR	179 (non-CRRT: 121, CRRT: 58)	Peak and trough
	Aréchiga-Alvarado NA [19]	2020	Prospective (TDM)	Critically ill mexican patients with suspected or proved infectious under treatment with amikacin	50 (45/5)	33.5 (18.0–64.0) ^b^	70.0 (44.0–138.0) ^b^	170.1 ± 7.9	24.0 (16.0–38.2) ^b^	Once daily IV dosing	1000 (500–1000) mg ^c^	2	80	0.5 and 12
	Petitcollin A [20]	2016	Prospective pharmacokinetic study	Ventilated critically ill patients on high-dose nebulized amikacin	20 (18/2)	57 (20–80)	67 (50–84)	NR	NR	20 mg/kg infusion of amikacin followed by either 3 other infusions or 3 nebulizations of 60 mg/kg amikacin (q24 h)	NR	33 (11–45) ^b^	522	0.5, 1, 1.5, 2, 3, 4, 6, 10, and 24
	French MA [21]	1981	Prospective and retrospective (TDM)	Critically ill patients	25 (15/10)	58 ± 14	NR	NR	NR	9 to 15 mg/kg per day	40.60 ± 42.67	NR	NR	NR
Gentamicin	Hodiamont CJ [22]	2017	Retrospective (TDM)	Critically ill patients on or off CVVH	44 (20/24)	61 (20–78)	70.5 (42.0–116)	170 (154–195)	NR	Starting dose of 4 mg/kg TBW, except for patientstreated for endocarditis due to Gram-positivemicro-organisms who were treated with 3 mg/kg in combination with a cell-wall-targeting antibiotic	4.0 (2.0–6.6) ^b^	NR	303	0.5 and the second sample was collected the next morning at 06:00 a.m., regardless of the time the first dose was administered
	Teigen MM [23]	2006	Prospective and retrospective (TDM)	Patients on hemodialysis receiving gentamicin to treat a suspected or proven infection	46 (23/23)	57.3 ± 17.3 (18–83)	72.4 ± 17.2 (42.1–100.5)	164.7 ± 11.6 (135–195)	NR	NR	NR	4.6 ± 2.2 (1–10)	NR	0.5, 1 sample at the beginning of dialysis, 1 sample at the end of dialysis, and 1 interdialytic blood sample taken prior to the next dialysis session
	Rea RS [24]	2008	Retrospective (TDM)	Critically ill patients	102 (45/57)	61.4 ± 16.8 (18.4–92.3)	81.4 ± 30.3 (29.0–222.3)	NR	NR	7 mg/kg/day	NR	2.1 (1–9)	211	NR
	Bos JC [25]	2019	Prospective observational pharmacokinetic study	Severally ill non-ICU sub-Saharan African Adult patients	48 (24/24)	40 (20–86)	51 (33–76)	NR	NR	80 to 160 mg/kg q8 h or 80 to 240 mg/kg q12 or 24 h	NR	NR	141	Predose, 30 to 120 min after intravenous administration and two random time points during the dosing interval
	Hodiamont CJ [26]	2017	Prospective (TDM)	Critically ill patients	59 (30/29)	60.9 ± 17.2	79.2 ± 22.0	NR	NR	Fixed first dose of approximately 5 mg/kg. Patients who were treated for endocarditis with 3 mg/kg in combination with a beta-lactam antibiotic	5.1 ± 1.1	6.7 ± 5.9	416	Peak and random timepoint between 6 and 23 h after the administration
	Roberts JA [27]	2010	Prospective pharmacokinetic study	Critically ill patients with acute kidney injury necessitating extended daily diafiltration	14 (13/1)	66.0 (57.0–74.5) ^b^	92.5 (80.0–111.1) ^b^	NR	NR	NR	NR	NR	265	0, 0.25, 0.5, 1, 2, 3, 5, 8, and 10
	Barletta JF [28]	2000	Prospective (TDM)	Critically ill trauma patients	19	40 ± 17 (17–75)	Adjusted (dosing) weight: 73.7 ± 15.9	NR	NR	NR	Gentamicine: 6.9 ± 0.39 (6–7.2) Tobramycine: 6.6 ± 1.03 (4.9–7.8)	NR	53	4 and 8
	Gomes A [29]	2017	Retrospective (TDM)	Endocarditis patients	65 (21/44)	69.3 (32–92)	76.2 (46–121)	173.9 (149–193)	NR	3 mg/kg q24 h	NR	NR	221	NR
	Watling SM [30]	1993	Prospective (TDM)	Critically ill patients	36 (20/16)	54.7 ± 16.6	75.7 ± 16.4	172 ± 15	NR	3 mg/kg q12 h, q18 h, q24 h, q36 h, or q72 h	NR	2.8 ± 1.6	102	1 h and at the dosing interval midpoint
	Kisor DF [31]	1992	Retrospective (TDM)	Patients with indicators of malnutrition (bodyweight less than ideal bodyweight, low serum ALB)	17 (16/1)	73.8 ± 11.8	54.3 ± 9.9	NR	NR	NR	NR	8.0 ± 1.2	72	NR
	French MA [21]	1981	Prospective and retrospective (TDM)	Critically ill patients	25 (15/10)	62 ± 15	NR	NR	NR	3 to 5 mg/kg per day	31.73 ± 27.26	NR	NR	NR
Tobramycin	Conil JM [32]	2011	Retrospective (TDM)	Critically ill patients	32 (27/5)	62.5 ± 15.3	77.5 ± 18.8	NR	NR	5 mg/kg q24 h for 3–5 days	NR	NR	NR	Peak and trough
	Aarons L [33]	1989	Retrospective (TDM)	Unselected poplation of patients treated with tobramycin	97 (52/45)	50.6 ± 19.0 (51.0;16–85) ^c^	66.5 ± 12.5 (66.8; 42–120) ^c^	NR	NR	NR	NR	(1–9)	322	2, 6 h after the dose for patients with normal renal function 2, 6, 12, and 24 h for patients with impaired renal function
	Hennig S [34]	2013	Retrospective (TDM)	Patients with or without cystic fibrosis	208 (109/99)	31.7 (18.0–85.0)	58.0 (37.0–120.0)	NR	NR	NR	5.2 (0.9–12.0) per day	NR	CF: 4514 No CF: 1095	NR

ALB, albumin; BMI, body mass index; CVVH, continuous venovenous hemofiltration; CVVHDF, continuous venovenous hemodiafiltration; ICU, intensive care unit; OA, open abdomen; NPT, negative pressure therapy; NR, not reported. ^a^ Values are expressed as the mean ± standard deviation (range) [interquartile range]. ^b^ Values are expressed as the median (range) [interquartile range]. ^c^ Values are expressed as the mean ± standard deviation (median; range).

**Table 2 antibiotics-10-00507-t002:** Population pharmacokinetic modeling methods and techniques used by the studies included in the review.

Drug	Study	Modeling			Simulation	
		Software	Model	Evaluation	Optimal Dosing Regimen	Target
Amikacin	Boidin C [16]	NPAG (Pmetrics)	2 compartments	Advanced internal	Optimal initial amikacin dose for C_max_: 3.5 g Optimal initial amikacin dose for AUC_0–24_: 3.8 g Optimal doses were based on an MIC of 8 mg/L	C_max_/MIC ≥ 8, AUC_0–24_/MIC ≥ 75 and C_min_ ≤ 2.5 mg/L
	Roger C [17]	NPAG (Pmetrics)	2 compartments	Advanced internal (bootstrap, *n* = 1000)	25 mg/kg every 48 h in critically ill patients receiving CRRT based on an MIC of 8 mg/L	C_max_/MIC ≥ 8 and C_min_ ≤ 2.5 mg/L
	Carrié C [18]	Monolix	2 compartments	Advanced internal (NPDE)	25–30 mg/kg every 36–48 h based on an MIC of 8 mg/L	C_max_/MIC ≥ 8, AUC_0–24_/MIC ≥ 75 and C_min_ ≤ 2.5 mg/L
	Aréchiga-Alvarado NA [19]	NONMEM 7.3	1 compartment	Advanced internal (bootstrap, *n* = 1000) and external (13 patients)	Based on an MIC of 8 mg/L and a dose of 30 mg/kg, the probability of having Cmax/MIC ≥ 8 was above 75% for creatinine clearance ranging from 60 mL/min to 200 mL/min ^a^	C_max_/MIC ≥ 8 and AUC_0–24/_MIC ≥ 75
	Petitcollin A [20]	Monolix 4.2.3	2 compartments	Advanced internal (NPDE)	–	–
	French MA [21]	NONLIN	2 compartments	NR	–	–
Gentamicin	Hodiamont CJ [22]	NONMEM 7.1.2	2 compartments	Advanced internal (bootstrap, *n* = 1000)	–	–
	Teigen MM [23]	NONMEM 5	1 compartment	Basic internal	Predialysis administration of 300 mg, 240 mg, and 220 mg as first, second, and third dose, respectively, for patients who dialyze 3 times a week	C_max_ ≥ 8 mg/L AUC_min,48h_ ≥ 140 AUC_max,48h_ ≤ 240
	Rea RS [24]	NONMEM 5.1	1 compartment	Advanced internal (bootstrap, *n* = 1000)	Initial doses of 7 mg/kg of either gentamicin or tobramycin. Then, it is recommended to verify Cmax after the first dose and determining MIC for the pathogen(s) with adjustment of subsequent doses to achieve the PD target ^b^	C_max_/MIC ≥ 10
	Bos JC [25]	NONMEM 7.1.2	1 compartment	Advanced internal (bootstrap, *n* = 1000)	7 mg/kg/day considering an MIC of 2 mg/L	C_max_/MIC ≥ 8
	Hodiamont CJ [26]	NONMEM 7.2	2 compartments	Advanced internal (bootstrap, *n* = 1000)	6 mg/kg as starting dose	C_max_ therapeutic range of 15–20 mg/L
	Roberts JA [27]	NONMEM 6.1	2 compartments	Advanced internal (bootstrap, *n* = 1000)	6 mg/kg every 48 h before the commencement of EDD-f	C_max_ > 10 mg/L and 70 mg·h/L ≤ AUC_0–24_ ≤ 120 mg·h/L
	Barletta JF [28]	Nonlinear mixed effect modelling	1 compartment	NR	–	–
	Gomes A [29]	MwPharm	1 compartment	Advanced internal (bootstrap, *n* = 1000) and external (14 patients)	–	–
	Watling SM [30]	NPEM ^c^	1 compartment	External of dosing nomogram only (15 patients)	–	–
	Kisor DF [31]	NPEM	1 compartment	NR	–	–
	French MA [21]	NONLIN	2 compartments	NR	–	–
Tobramycin	Conil JM [32]	NONMEM 5	2 compartments	Advanced internal (NPDE and bootstrap, *n* = 1000) and external (17 patients)	Peak and AUC pharmacodynamic targets could not be reached simultaneously in more than 45% of the ICU patient population. Combination therapy in addition to TDM are required to manage efficacy and toxicity	C_max_/MIC > 10, C_min_ ≤ 1 mg/L AUC between 80 and 125 mg·h/L for MIC ≤ 1 mg/L
	Aarons L [33]	NONMEM	2 compartments	External (34 patients)	First 48 h: 100 mg Q8 h and Maintenance dose: 120 mg Q8 h, patient with CLcr > 100 mL/min First 48 h: 80 mg Q8 h and Maintenance dose: 90 mg Q8 h, patient with CLcr = 75 mL/min First 48 h: 93 mg Q12 h and Maintenance dose: 90 mg Q12 h, patient with CLcr = 50 mL/min First 48 h: 60 mg Q12 and Maintenance dose: 54 mg Q12 h, patient with CLcr = 30 mL/min First 48 h: 80 mg Q24 and Maintenance dose: 70 mg Q24 h, patient with CLcr = 20 mL/min First 48 h: 67 mg Q24 and Maintenance dose: 54 mg Q24 h, patient with CLcr = 15 mL/min First 48 h: 60 mg Q24 and Maintenance dose: 35 mg Q24 h, patient with CLcr = 10 mL/min	C_max_ = 6 mg/L and average concentrations within a dosing interval ≤ 4 mg/L
	Hennig S [34]	NONMEM 7.2	2 compartments	Advanced internal (bootstrap, *n* = 300)	11 mg/kg/day for Cystic Fibrosis patients	C_max_ = 20 mg/L (relating to a 1-h peak/MIC ratios of 20/2) and C_min_ < 1 mg/L

AUC, area under the concentration–time curve; CLcr, creatinine clearance; C_max_, maximum concentration; C_min_, minimum concentration; CRRT, continuous renal replacement therapy; ICU, intensive care unit; MIC, minimal inhibitory concentration; NPDE, normalized prediction distribution error; NR, not reported. ^a^ Graphical representation of probability of target attainment based on different amikacin dosing regimens (15 mg/kg to 70 mg/kg), different MIC (4 mg/L to 16 mg/L), and different values of creatinine clearance. ^b^ Table probability of C_max_ ≥ 10 × MIC by different MIC and aminoglycoside dose. ^c^ PK parameters were calculated using Sawchuk–Zaske method.

## Data Availability

Not applicable.

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
