# Peer review of "Aminoglycosides in the Intensive Care Unit: What Is New in Population PK Modeling?"

_antibiotics, 2021, doi:10.3390/antibiotics10050507_

Round 1

Reviewer 1 Report

Duong et al. describe in their review on “Aminoglycosides in the intensive care unit: What's new in population PK modeling?” previously published PK models, the structure of these models and recommended dosing regimens. The authors of the study are to be congratulated on this fine work. Overall, there are only minor comments:

  • The review has become very long overall. I would recommend the authors to critically review which parts could be moved to the appendix and where you can also significantly shorten the text to increase reader-friendliness. Some suggestions:
    • Table 3 should be moved to the Appendix.
    • The discussion section on individual covariates is extensive, I recommend to shorten here clearly (section 4.2. and 4.3 are nice).
  • How about the development of a meta-model with inclusion of all available datasets? I would like to read about this in the discussion
  • It is nice that you mention external validation. You could focus on this a bit more, e.g. suggest strategies how to evaluate the models (e.g. by a retrospective dataset using a simulation approach of the available models, which is quite simple, but would be very helpful to assess the predictive performance of the models).
  • Figures are not referenced in the text correctly, please remove the automated references and include manual references.
  • How about inter- and extrapolation of the models (based on the respective study populations). It would be nice to discuss this.

Author Response

  • Table 3 should be moved to the Appendix.

Thank you for your suggestion. Following review, we also agree that table 3 (significant covariates) should be moved to the Supplementary Information (now labeled as Table S4).

  • The discussion section on individual covariates is extensive, I recommend to shorten here clearly (section 4.2. and 4.3 are nice).

Upon reviewing the sections on individual covariates, we decided to keep the first two paragraphs: 4.1.1-Renal Function and 4.1.2.-Bodyweight and bodysize since they were the most frequent significant covariates retained in the final population pharmacokinetic models across all three aminoglycosides. We also decided to move a paragraph from 4.1.3-Age and add to section 4.1.1-Renal function since it covered the impact of age on renal function.

  • How about the development of a meta-model with inclusion of all available datasets? I would like to read about this in the discussion

Thank you for this suggestion, the possibility of developing a meta-model was discussed in section 4.2 of the discussion. Moreover, it is important to note that this current review also falls within another project where we plan to evaluate these models from the literature with independent datasets from two hospitals established in the province of Quebec, Canada.

  • It is nice that you mention external validation. You could focus on this a bit more, e.g. suggest strategies how to evaluate the models (e.g. by a retrospective dataset using a simulation approach of the available models, which is quite simple, but would be very helpful to assess the predictive performance of the models).

Although this review appears to be relatively lengthy, we agree that the different strategies of external evaluation should be described. Therefore, a small section of these different steps and strategies were added in the Supplementary Information.

  • Figures are not referenced in the text correctly, please remove the automated references and include manual references.

Cross-references hyperlinks across this manuscript were functional and adequate before submission to the editor. The modifications made by the editor might have made these links unusable. This issue should be fixed during finalization.

  • How about inter- and extrapolation of the models (based on the respective study populations). It would be nice to discuss this.

Details were added on the different articles that performed inter- and extrapolation of the simulated dosing regimens based on their respective actual dose administered in section 4.3. Results based on these inter and extrapolation should be interpreted cautiously due to the high variability observed in ICU patients.

Reviewer 2 Report

The paper deals with the pharmacokinetics of aminoglycosides in the intensive care unit.

The paper is well written and well designed. I recommend its publication, after minor revision, according to the following points:

  1. Figure 1: it is not clear the reason for the exclusion of 5 records in the second step; I suggest to explicate it
  2. table 1: it is at the moment subdivided into three parts on the same page; I think it is a redactional problem
  3. section 3.4: the Authors reports the use of both single- and two-compartment models; although this question depend on the original papers, I think it is relevant that the readers know the reasons of these different types of models
  4. Abstract: for the reasons expressed in the previous remark (3.) I suggest that the expression "Two-compartment model best described" (raw 23) should be substituted by "Two-compartment model was used to describe"
  5. Figure 2a, 2b: I suggest adding the results of an ANOVA to compare Clearance and Volume among the Aminoglycosides, to complete the reported information (followed by a post hoc comparison test)
  6. Figure 2a, 2b:  If it is possible, I suggest building a box and whisker plot also for healthy volunteers (instead of a single line, as actually is); in this case, I suggest using the Dunnett test as a post hoc comparison test

Author Response

  1. Figure 1: it is not clear the reason for the exclusion of 5 records in the second step; I suggest to explicate it

First exclusion step (n=54) was due to the exclusion criteria while the second exclusion step was due to inaccessibility of the full-text article. The reason of both exclusion steps is now defined in the Figure 1 itself.

  1. table 1: it is at the moment subdivided into three parts on the same page; I think it is a redactional problem

Upon submission to the editor, this table was not split into three parts.This issue should be fixed during finalization.

  1. section 3.4: the Authors reports the use of both single- and two-compartment models; although this question depend on the original papers, I think it is relevant that the readers know the reasons of these different types of models

Section 3.4 consists of a presentation of the results in a descriptive matter. Possible reasons explaining these different types of models (single- and bi-compartment models) were described in Section 4 of the discussion (Lines 261-267) as follows :

“The 19 articles presented in this review exhibit many resemblances but also differences on the covariates included, the structure of the model and the simulation of dosing regimens. Studies presenting a design with TDM samples or a sparse sampling schedule were mostly associated with single-compartment models (n=8), whereas full profile sampling partially led to bi-compartment models (n=11). In fact, Marsot et al. suggested in their review that single-compartment models could lead to an inaccurate estimation of aminoglycosides Vd [15]

  1. Abstract: for the reasons expressed in the previous remark (3.) I suggest that the expression "Two-compartment model best described" (raw 23) should be substituted by "Two-compartment model was used to describe"

We agree with this recommendation and this sentence was corrected as per suggestion.

  1. Figure 2a, 2b: I suggest adding the results of an ANOVA to compare Clearance and Volume among the Aminoglycosides, to complete the reported information (followed by a post hoc comparison test)

Anova test was performed for both CL and Vd across all three aminoglycosides. Mean clearance and Vd were statistically not different among the aminoglycosides (p> 0.05).

Anova results was included as footnote in the updated figures 2a and 2b.

  1. Figure 2a, 2b:  If it is possible, I suggest building a box and whisker plot also for healthy volunteers (instead of a single line, as actually is); in this case, I suggest using the Dunnett test as a post hoc comparison test

Thank you for this suggestion. Although we also agree that it would have been ideal to present both clearances and volume of healthy volunteers in a form of a boxplot, only mean values were presented in the reported literature.

Considering that the Dunnett test is applicable when the control/theorical group contains a list of values, it wouldn’t be feasible in this context. Therefore, as an alternative, we decided to perform a Student t-test for each aminoglycoside. For each parameter and each aminoglycoside, values in critically ill patients were tested against their respective mean value in healthy volunteer.

The results (p>0.05 or p<0.05) for each aminoglycoside are presented under their respective labels in figure 2a and 2b.

Reviewer 3 Report

The topic of the manuscript is interesting. The reviewer feels the manuscript can be accepted after some minor amendments.

(1) Abstract:

Method: ... Please check spelling

(2) Why the authors only review the publications from ICU? Why the literature outside ICU is not included?

(3) Any influence from pharmacogenetics ?

Author Response

(1) Abstract:

Method: ... Please check spelling

Spelling in the method section of the abstract was reverified.

(2) Why the authors only review the publications from ICU? Why the literature outside ICU is not included?

As per our objective, this literature review only targeted ICU patients considering the known high variability in terms of aminoglycosides dosing regimens administered and pharmacokinetics. This review also falls within a global project where we plan to evaluate these models from the literature with independent datasets from two hospitals established in the province of Quebec, Canada.

(3) Any influence from pharmacogenetics ?

Thank you for this comment. Considering that aminoglycosides-induced ototoxicity is reported to be more frequent amongst individuals with mitochondrial DNA mutations, it should be considered in future development of population pharmacokinetics model. Details were added in the conclusion.